# Radionuclide Imaging of Cytotoxic Immune Cell Responses to Anti-Cancer Immunotherapy

**DOI:** 10.3390/biomedicines10051074

**Published:** 2022-05-05

**Authors:** Louis Lauwerys, Evelien Smits, Tim Van den Wyngaert, Filipe Elvas

**Affiliations:** 1Molecular Imaging Center Antwerp (MICA), Integrated Personalized and Precision Oncology Network (IPPON), Faculty of Medicine and Health Sciences, University of Antwerp, Universiteitsplein 1, B-2610 Wilrijk, Belgium; louis.lauwerys@uantwerpen.be (L.L.); tim.vandenwyngaert@uantwerpen.be (T.V.d.W.); 2Center for Oncological Research (CORE), Integrated Personalized and Precision Oncology Network (IPPON), Faculty of Medicine and Health Sciences, University of Antwerp, Universiteitsplein 1, B-2610 Wilrijk, Belgium; evelien.smits@uantwerpen.be; 3Center for Cell Therapy and Regenerative Medicine, Antwerp University Hospital, Drie Eikenstraat 655, B-2650 Edegem, Belgium; 4Nuclear Medicine, Antwerp University Hospital, Drie Eikenstraat 655, B-2650 Edegem, Belgium

**Keywords:** immunotherapy, biomarkers, molecular imaging, in vivo imaging, positron emission tomography, T cell imaging, cell imaging, oncology, activation markers, exhaustion markers

## Abstract

Cancer immunotherapy is an evolving and promising cancer treatment that takes advantage of the body’s immune system to yield effective tumor elimination. Importantly, immunotherapy has changed the treatment landscape for many cancers, resulting in remarkable tumor responses and improvements in patient survival. However, despite impressive tumor effects and extended patient survival, only a small proportion of patients respond, and others can develop immune-related adverse events associated with these therapies, which are associated with considerable costs. Therefore, strategies to increase the proportion of patients gaining a benefit from these treatments and/or increasing the durability of immune-mediated tumor response are still urgently needed. Currently, measurement of blood or tissue biomarkers has demonstrated sampling limitations, due to intrinsic tumor heterogeneity and the latter being invasive. In addition, the unique response patterns of these therapies are not adequately captured by conventional imaging modalities. Consequently, non-invasive, sensitive, and quantitative molecular imaging techniques, such as positron emission tomography (PET) and single-photon emission computed tomography (SPECT) using specific radiotracers, have been increasingly used for longitudinal whole-body monitoring of immune responses. Immunotherapies rely on the effector function of CD8^+^ T cells and natural killer cells (NK) at tumor lesions; therefore, the monitoring of these cytotoxic immune cells is of value for therapy response assessment. Different immune cell targets have been investigated as surrogate markers of response to immunotherapy, which motivated the development of multiple imaging agents. In this review, the targets and radiotracers being investigated for monitoring the functional status of immune effector cells are summarized, and their use for imaging of immune-related responses are reviewed along their limitations and pitfalls, of which multiple have already been translated to the clinic. Finally, emerging effector immune cell imaging strategies and future directions are provided.

## 1. Introduction

In recent years, a shift has been made from traditional chemotherapy and radiotherapy to targeted treatments and immunotherapy for cancer treatment, including the clinical use of immune checkpoint inhibitors (ICI), adoptive cell therapy and cancer vaccines with over 3000 ongoing clinical trials [1,2,3,4]. Cancer immunotherapies generally work by promoting cancer cell death through enhancing cytotoxic immune cell infiltration and function, and by reprogramming myeloid cells toward an anti-tumor phenotype [5,6]. Despite the impressive responses seen in patients with rapidly fatal cancers, only a fraction of patients shows a durable response to immunotherapy and a significant number of patients develop severe adverse events [1,7]. Furthermore, immunotherapy is expensive, leaving fewer resources available for other purposes [8]. Possible reasons for the lack of response in some patients are complex interactions in the tumor microenvironment (TME), physical barriers that prevent lymphocyte infiltration, feedback loops and checkpoints in the immune response, and the heterogeneity and adaptability of tumors [9]. Therefore, it would be of value for physicians to quickly identify responders from non-responders to optimize patient treatment, minimize severe adverse events, and reduce inefficient usage of healthcare resources.

The effect of immunotherapy relies on enhancing the immune response of cytotoxic immune cells to cancer as described by the cancer-immunity cycle [10,11]. After internalizing an antigen at the tumor site, antigen presenting cells (APCs) migrate to secondary lymphoid organs (SLOs), where they present the antigen to T cells. This process is followed by T cell antigen recognition, activation, and subsequent population expansion, whereafter the activated T cells depart from the SLOs and home towards the tumor. This T cell activation and population expansion is characterized by a metabolic switch, with increased glucose uptake and an increased usage of deoxyribonucleoside salvage pathways. The final step before T cells execute their cytotoxic effector function is to infiltrate the TME. In the case of an effective anti-cancer immune response, the resulting tumor cell death and inflammation due to T cell cytotoxic activity leads to new waves of effector cells [6].

Most often, the eligibility of patients to receive immunotherapy is determined using tumor biopsies to characterize the TME in an attempt to predict response. However, biopsies are not always representative of the heterogeneity and dynamic change in the TME [12,13]. In addition, taking biopsies is an invasive procedure, which is generally undesired [12,13]. As an alternative, liquid biopsies are proposed as a powerful tool to distinguish between subpopulations of cancer cells. A disadvantage of this technique is that liquid biopsies are not suitable for disease staging, due to a lack of spatial information and detail on tumor heterogeneity [12,14]. After therapy is initiated, evaluation of tumor response is conventionally performed in clinical practice using the RECIST criteria based on changes in tumor dimensions, as measured by computed tomography (CT) or magnetic resonance imaging (MRI) [15,16]. However, with immunotherapies, these anatomic imaging techniques cannot differentiate between tumor progression and pseudo-progression, a state where tumor volume increases due to inflammation and infiltration of immune cells, followed by an eventual decrease in total tumor burden [17,18]. Even though modifications to the RECIST system have been developed to overcome some of these limitations (e.g., iRECIST), morphological tumor changes are only visible after multiple weeks, which is unfavourable for patients, due to disease progression and adverse therapy effects [19]. Therefore, clinicians need other tools to predict and evaluate treatment responses more accurately.

Molecular imaging with positron emission tomography (PET) or single-photon emission computed tomography (SPECT) are non-invasive techniques that can monitor cellular processes after injection of tracer amounts of radiolabelled molecules (also known as radiopharmaceuticals) [20]. Molecular imaging can be used to visualize the homing and accumulation of cytotoxic cells at tumor sites using tracers specific for certain targets, such as CD8, which can give an indication of tumor infiltration [21]. Different methodologies have been explored to radiolabel immune cells, each with their own benefits and drawbacks [22]. Immune cells can either be labelled with imaging probes before administration, modified with a reporter gene ex vivo, or visualized directly in vivo with target-specific probes [22]. However, challenges remain in measuring response to therapy because the TME is immunosuppressive and downregulates the activity of infiltrated cytotoxic immune cells in different ways [23]. Whereas activated immune cells show increased expression of various targets (e.g., NKp30 on natural killer cells [NK]), exhausted immune cells (i.e., cytotoxic immune cells with an impaired effector function) stop expressing specific markers (e.g., granzyme B) [24,25]. Quantification of expression of activation markers could therefore provide unique insights in the treatment response to immunotherapy compared to quantifying the accumulation of lymphocytes. Moreover, ICI treatment response can be predicted by measuring the co-expression of inhibitory receptors, such as LAG3 and PD-1, both cell exhaustion markers, on tumor infiltrating lymphocytes (TILs) [26,27]. Therefore, evaluating the functional state of cytotoxic cells by imaging activation or exhaustion markers may provide information on response to immunotherapy at an earlier stage than morphological methods, benefiting patient management [19,28,29].

In the first part of this paper, an overview is provided of the different radiotracers used to image total immune infiltrates. The main part of this review discusses recent advances made in evaluating the functional state of effector immune cells in the TME using PET and SPECT imaging to predict or determine response to immunotherapy.

## 2. Imaging of Total Immune Infiltrates 

The core concept of adoptive cell therapies, such as CAR T cells and CAR NK cells, is to insert a chimeric antigen receptor (CAR) in extracted lymphocytes for recognition of tumor antigens after reinsertion in the body [30]. Moreover, immunotherapies rely on the activation, proliferation, tumor infiltration and subsequent cytotoxic effector function of immune cells [11,31]. Therefore, knowing the extent of lymphocyte accumulation at the tumor site is a good biomarker for therapy response [11,31]. Three different methods for cell labelling with radiopharmaceuticals and subsequent tracking, have been explored, each with their own pros and cons. To assess the effect of adoptive immune cell transfer therapy, cells can be labelled directly ex vivo with a suitable radiotracer for in vivo imaging [32]. An alternative is inserting reporter genes in the immune cells that program proteins which radiopharmaceuticals can specifically target [33,34]. A final method to assess immune response is by imaging lymphocytes in vivo by targeting markers that are constitutively expressed on the cells of interest [35,36]. The three different approaches are discussed in the next paragraphs (Figure 1). 

Direct ex vivo labelling of immune cells consists of labelling of cells with an agent that remains trapped within the cells, followed by injection of the cells to the patient. This technique is commonly used in clinical practice as it does not require the modification of the cells. [^111^In]In-oxine, [^89^Zr]Zr-oxine and [^18^F]FDG have been used for this purpose among others (Table 1) [37,38,39]. However, a major drawback of this method is that the tracer is diluted with every cell division, diminishing the measurable signal per cell [32]. Additionally, the efflux of loaded radiotracers decreases the measurable signal and increases background radiation, resulting in higher amounts of activity administration and accompanying increased patient radiation burden. This efflux is partially mitigated by a more recent tracer, [^89^Zr]Zr-DFO-Bz-NCS, which forms a biostable covalent thiourea bond between the amine functionalities and the NCS group. However, this radiotracer suffers from the known issue of instability of the ^89^Zr-DFO metal complex. In order to increase the stability of ^89^Zr complexes two DFO chelator analogs have been developed and are currently under clinical evaluation [38,40]. Moreover, toxicity to the isolated cells, patient and manipulator must be taken into account when using the direct ex vivo cell labelling method, and loss of cell function must be tested extensively [32]. 

An alternative is to use indirect labelling approaches that do not suffer from loss of labelling agent from the cells and tracer dilution. These indirect labelling strategies refer to the use of reporter genes, or to the in vivo targeting of specific antigens expressed by immune cell subtypes.

Reporter gene imaging consists of introducing genes, such as *HSV1-TK*, in a cell of interest under the control of a given promotor. These genes are transduced in specific receptors, proteins or enzymes, that are then targeted after injection of a radio-labelled agent (Table 2). Since the gene can be passed with cell division, imaging can be performed at late time points by repeated injection of the radio-labelled agent, and the fate of the cells or their progeny can be followed until they die. The method is associated with a lower cytotoxicity than the direct ex vivo labelling method. However, the approach still suffers some limitations [32]. Firstly, special attention has to be taken to ensure no loss of cell function occurs due to the gene editing, and immunogenicity has to be avoided [32]. Furthermore, this approach has a significantly higher cost than the direct labelling method, and specialized equipment and training is necessary [32]. Finally, the clinical implementation of reporter gene-based cell-tracking methods comes with an important regulatory burden, which explains why clinical studies that use reporter genes for immune cell imaging have yet to start [32]. However, recent developments in gene-editing strategies make it possible to insert the desired reporter gene in safe harbor locations, making safe use of this method easier [48,49].

Other indirect approaches rely on specific radio-labelled monoclonal antibodies (mAbs) or peptides that bind to target antigens expressed on the plasma membrane of different cell subsets. An approach that avoids the need to manipulate immune cells ex vivo, involve administering radiotracers that can selectively identify endogenously expressed markers of immune cells in vivo (Table 3) [21]. This approach suffers less cytotoxicity problems than the direct loading method, and clinical translation is less troublesome, from a regulatory point of view, than reporter gene methods [32]. However, contrary to the direct labelling method, the tracers are subject to biodistribution effects of the body and high background signals can pose interpretation difficulties. The use of smaller constructs such as mini-bodies and peptides results in faster clearance and accompanying lower background than larger tracers, such as mAbs [76]. The potential of this approach is demonstrated by Rashidian and colleagues, who administered the CD8-targeting PET tracer [^89^Zr]Zr-PEG20-x118-VHH to wild type mice grafted with a B16 melanoma. The imaging results showed that strong therapy responders had a homogenous PET signal in tumors, due to CD8^+^ T cell infiltration, whereas weak responders had a more heterogenous signal distribution in the tumor [77]. 

## 3. Imaging of Markers of Activated Effector Cells

A limitation of the previous methods is lack of information on the functional status of cytotoxic immune cells. Therefore, the actual therapeutic effect of immunotherapy might be overestimated in patients where the function of the immune cells is downregulated, which could result in tumor progression. As a result, a large amount of research has been done in developing ways to assess treatment response more accurately. To this purpose, different tracers have been developed to image markers that are overexpressed or released by activated effector immune cells (e.g., CTLA-4 and granzyme B) to differentiate between active immune cells and exhausted immune cells [24,91,92]. Alternatively, the imaging of the metabolic switch associated with immune cell activation is another potential method to determine therapy response [93,94]. By focussing on active immune cells, rather than total tumor immune infiltrate, imaging biomarkers of cytotoxic immune cell activity may be more valuable for predicting response to cancer immunotherapy. The different tracers developed to target these markers are discussed in the following paragraphs (Figure 2).

### 3.1. Probes for Cell Surface Markers and Secreted Molecules

#### 3.1.1. Granzyme B

Granzyme B is a serine protease stored in secretory vesicles in NK cells and CD8^+^ T cells that mediates cancer cell death after cell activation and sequential release from the vesicles [24]. Monitoring the secretion of granzyme B would provide a good indication of therapeutic response to immunotherapies, as this represents a major mechanism responsible for immune-mediated cell death [95]. 

Larimer et al. developed a granzyme B-targeted PET activity-based probe (GZP), bearing the granzyme B recognition sequence, Ile-Glu-Phe-Asp, attached to an aldehyde warhead. This ABP contained the radiometal chelator NOTA for radio-labelling with ^68^Ga yielding [^68^Ga]Ga-NOTA-GZP [24,96]. The in vitro results indicated that the tracer has high specificity for granzyme B over other granzyme family members, yet selectivity over closely related proteases was not demonstrated [24,97]. The tracer was tested in mice xenografted with CT26 cancer cells, which showed that the peptide is eliminated renally and that the uptake in the tumor is predictive for therapy response with anti-PD-1 and anti-CTLA-4 combination therapy [24]. In a large in vivo study, GZP accumulation in the tumor had an overall positive predictive value of 84% for anti-PD-1 and anti-CTLA-4 immunotherapy response in the murine colon cell lines CT26 and MC38-tumor- bearing mice (N=31) and a negative predictive value of 94% (N=35) [98]. Goggi and colleagues developed a [^18^F]AlF analog for the same IEFD recognition sequence, [^18^F]AlF-mNOTA-GZP (NOTA–β-Ala–Gly–Gly–Ile–Glu–Phe–Asp–CHO), that showed similar predictive results in CT26 and MC38-tumor-bearing mice as [^68^Ga]Ga-NOTA-GZP [92]. A distinct advantage of using ^18^F-tracers, instead of ^68^Ga-based tracers, is the improved imaging quality associated with ^18^F, and, in addition, the radiation burden is lower despite having a longer half-life, which is also preferable [74,99]. However, ^68^Ga is eluted from a generator, whereas ^18^F production requires a cyclotron, limiting its usage in facilities lacking the necessary equipment, in addition to the higher cost of production [74,99]. 

Although the structure and sequence of human granzyme B and mouse granzyme B are similar, their substrate specificity is different with human granzyme B preferring the tetrapeptide sequences IEPD, IETD or IEQD over the IEFD-sequence that is recognized specifically by mouse granzyme B, as determined by Casciola-Rosen and colleagues using positional scanning synthetic substrate combinatorial libraries [100]. Therefore, the same group developed an analogous biotinylated probe for human granzyme B by changing one amino acid of the tetrapeptide to create hGZP, which showed high specificity to human granzyme B in human melanoma biopsies [24]. Although the IEPD sequence can also be recognized by other proteases, such as caspase-8, the fact that granzyme B is released into the extracellular space from activated T cells, makes this protease preferentially accessible to binding by the imaging agent over intracellular counterparts [24,101].

Recently, a restricted interaction peptide (RIP) that consists of a labelled antimicrobial peptide fragment that integrates in the cell membrane, a cleavage sequence (recognized by granzyme B), and a peptide masking domain, was designed by Zhao et al. [102]. The group chose the octapeptide IEPDVSQV as cleavage site, due to the high specificity of IEPD for granzyme B over other human granzymes and the increased catalytic activity observed after addition of VSQV. Temporin L was used as the membrane interacting domain and the PAR1 peptide served as the peptide masking domain. To prove that it is the cleaved version of GRIB B that binds membranes, an in vitro assay demonstrated that membrane integration of [^64^Cu]Cu-GRIP B was significantly higher after co-incubation with granzyme B. Rapid renal excretion was demonstrated through biodistribution studies in nude BALB/c mice. A significantly higher tracer uptake was seen in the tumors and spleens of CT26 mice treated with anti-PD-1/anti-CTLA-4 than in a control group, and post-treatment tracer accumulation correlated with a decrease in tumor volume in wild-type mice treated with ICI [102]. One of the advantages that ^64^Cu-GRIP B has over the ^18^F and ^68^Ga probes is that the unique targeting method allows the imaging at a later time point (>4h), improving the contrast with background [101,102].

#### 3.1.2. IFN-γ

Interferon gamma (IFN-γ) is a cytokine secreted by CD4^+^ T cells, CD8^+^ T cells and NK cells that plays a major role in tumor cell recognition and destruction through different pathways [103]. Binding of IFN-γ with IFN-γ receptor leads to dimerization of STAT1 and translocation to the nucleus where it binds its target genes, which leads to a diverse response, including upregulation of major histocompatibility complex (MHC) on APC and upregulation of the Fas/FasL pathway [104]. 

Gibson and colleagues developed a mAb [^89^Zr]Zr-anti-IFN-γ PET tracer to detect elevated IFN-γ levels after cancer treatment with immunotherapy as a biomarker for immune activation in tumors [103]. The tracer was tested in neu^+^ TUBO tumor-bearing BALB/c mice treated with HER2/neu DNA vaccination. Significantly higher tracer accumulation was seen in the tumor of treated mice, which was ex vivo correlated to immune cell infiltration. The researchers were able to distinguish in situ between activated and exhausted T cells using the tracer in the aforementioned mouse model by correlating a decreased tracer accumulation with an increased PD-1 expression on CD8^+^ T cells [103].

#### 3.1.3. IL-2R

Interleukin-2 (IL-2) is a small glycoprotein secreted mainly by T cells during immune responses. Secreted IL-2 binds the IL-2 receptor (IL-2R), followed by lymphocyte proliferation, cytokine secretion and the expression of MHC II on T cells [105]. The IL-2R is expressed not only on activated cytotoxic T cells, but also on regulatory T cells. Therefore, quantitative imaging of IL-2R represents a good indicator of immune responses. 

D’alessandria and colleagues developed a radio-labelled IL-2 probe, [^99m^Tc]Tc-succinimidyl-6-hydrazinopyridine-3-carboxylate-IL2 ([^99m^Tc]Tc-HYNIC-IL2), using a single-step synthesis [106]. A clinical study with 30 melanoma patients demonstrated that the tracer is safe and is able to visualize T cell activation [107]. Another SPECT tracer, [^123^I]I-IL-2, was developed for the same purpose and a clinical study in seventeen carcinoma patients showed that the tracer allows the imaging of IL-2R on TILs [108]. However, PET-tracers are preferred over SPECT-tracers in the clinic, due to their higher sensitivity and spatio-temporal resolution [109]. Therefore, different groups have worked on developing PET-probes to image IL-2R expression.

In that regard, an IL-2 PET tracer N-(4-[^18^F]-fluorobenzoyl)interleukin-2 ([^18^F]FB-IL2) was developed by the De Vries group [105]. The final construct was created by adding purified [^18^F]-fluorobenzoate ([^18^F]FSB) to IL-2 and purifying the labelled glycoprotein using RP-HPLC [105]. Ex vivo biodistribution studies in mice showed that the radiotracer is mainly excreted through the renal pathway and that bone uptake is minimal, indicating negligible in vivo defluorination [105]. The tracer was injected in SCID immune-incompetent mice with increasing amounts of human T lymphocytes grafted in the right shoulder. The PET scan showed increased activity in the right shoulder but also a considerable amount of activity in the control shoulder, due to activated T cell migration to the inflamed tissue. No correlation was found between activity in the right shoulder and number of human T cells injected. However, total activity in both shoulders did correlate with the amount of human T cells [105]. 

A small cohort of patients was administered [^18^F]FB-IL2 before and six weeks after start of immune checkpoint inhibition therapy with ipilimumab and/or nivolumab. High uptake was noted in the lymphoid system and the major excretory organs (liver and kidneys). Three of the 11 patients showed high uptake in the lungs, possibly due to the formation of tracer micro-aggregates. The study demonstrated that [^18^F]FB-IL2 is safe to use in patients at 200 MBq, 50 µM. However, the activity at the lesion site was counter-intuitively lower after therapy than at baseline. A possible explanation that the researchers gave for this is that IL-2R is only found on regulatory T cells in tumor lesions with no significant change over time, which would mean that [^18^F]FB-IL2 does image IL-2R in the TME, but is not, however, a good biomarker for therapy response. Other possibilities for the low uptake are the formation of anti-drug antibodies (ADA) and low binding due to competition with endogenous IL-2 [110]. Another drawback for clinical use of this tracer is that when translating the three-step, partly manual radiosynthesis to a fully automated, GMP compliant method to use the tracer in clinical studies, yields only enough for two patients per production [105,111] were acquired.

Van der Veen and colleagues developed two other IL-2 tracers with shorter production times, [^18^F]AlF-RESCA-IL2 and [^68^Ga]Ga-NODAGA-IL2. The chelator RESCA, or restrained complexing agent, is a relatively novel chelator with improved properties for binding metals to heat-sensitive biologicals, due to reasonably fast reaction kinetics at room temperature [112]. Both developed tracers only use one synthesis module in contrast to the two necessary for [^18^F]FB-IL2, and [^18^F]AlF-RESCA-IL2 yields were sufficient for multiple patients, making it more feasible for clinical implementation. [^68^Ga]Ga-NODAGA-IL2, however, was produced in lower yields than [^18^F]AlF-RESCA-IL2 and production still has to be optimized before clinical use. Both tracers showed an in vitro stability over 90%. The in vitro uptake in activated hPBMCs of [^18^F]AlF-RESCA-IL2 (73 ± 26.3%) and [^68^Ga]Ga-NODAGA-IL2 (12.7 ± 0.1%) were higher than the one of [^18^F]FB-IL2 (4.8 ± 2.8%). In contrast to [^18^F]FB-IL2, no uptake was seen in the control shoulder in immunodeficient mice with activated human peripheral blood mononuclear cells (hPBMCs) in Matrigel grafts. No uptake was seen in either shoulder after co-injection with a high dose of IL-2, proving the specificity of the tracers [113]. Ex vivo biodistribution studies in immunodeficient mice showed a similar profile to that of [^18^F]FB-IL2, with increased uptake in the liver and kidneys. At this time, the clinical results of IL-2R imaging are difficult to interpret and additional tracer optimization must be performed before the clinical utility of these imaging agents is demonstrated.

#### 3.1.4. ICOS

The inducible T cell costimulatory receptor (ICOS), also known as CD28, decreases the activation threshold of T cells after interaction with CD80/B7.1 or CD86/B7.2 on APC and plays a regulatory role in adoptive T cell response [114]. ICOS is upregulated during T cell activation and has thus been identified as a potential biomarker for CAR T cell activation [115]. Interestingly, ICOS is not constitutively expressed on resting T cells, representing an exclusive biomarker for activated T cells [115].

Simonetta and colleagues studied [^89^Zr]Zr-DFO-ICOS mAb as an immuno-PET tracer for in vivo monitoring of CD19-specific CAR T cells for B cell lymphoma treatment in immuno-deficient BALB/c mice. Due to the large size of the antibody, [^89^Zr]Zr-DFO-ICOS mAb accumulated in well-vascularized organs, such as the heart, liver and spleen in untreated mice 48 h post-injection which restricts the detection of CAR T cells in the spleen. In the study, mice treated with CD19-specific CAR T cells showed a higher [^89^Zr]Zr-DFO-ICOS mAb uptake in bone marrow than both untreated mice and mice treated with untransduced T cells, which was indicative of higher ICOS expression in bone marrow. The higher ICOS expression was corroborated using flow cytometry analysis [115]. The administration of [^89^Zr]Zr-DFO-ICOS mAb showed no detectable change in CAR T cell proliferation and therapeutic effect, making it feasible for clinical translation [115].

#### 3.1.5. OX40

The TNF receptor superfamily member OX40 is transiently expressed by CD4^+^ and CD8^+^ T cells following TCR stimulation and is transiently expressed on NK cells as well [116]. Thus, antigen-specific activated T cells are potential targets of OX40-directed monitoring of immune activation [117].

Alam et al. developed a PET-tracer based on a murine OX40-specific monoclonal antibody, [^64^Cu]Cu-DOTA-OX40 mAb, to non-invasively image OX40+ activated T cells in vivo. The researchers demonstrated that the cell-binding affinity of the DOTA-AbOX40 was comparable to the cell-binding affinity of the unmodified AbOX40 in an in vitro activated murine T cell-binding experiment [117]. A murine dual-tumor lymphoma model was used to image CD4^+^OX40^+^ T cells in tumors vaccinated with CpG oligodeoxynucleotide. Using this model, the researchers were able to observe significantly increased uptake in the tumors and tumor draining lymph nodes of treated mice compared to untreated mice. The increased uptake could also be measured in the spleen and more distal lymph nodes on day 2 after tracer administration [117]. The same group developed [^89^Zr]Zr-DFO-OX40 mAb, which has a longer half-life than ^64^Cu, and evaluated it in a murine glioblastoma model treated with CpG oligodeoxynucleotide to observe longer evaluation of the kinetics of the whole-body immune response to the vaccination. The in vivo PET scans showed an increased activity at days 1, 2 and 5 in the lymph nodes of the vaccinated mice compared to the lymph nodes of control mice. An interesting observation is that there was a higher accumulation in the brain of the control mice than of the treated mice. The fact that lymph nodes further away from the vaccination site also showed OX40^+^ T cells indicate that the immune reaction initiated by CpG-vaccination can be analyzed in the whole body. The specificity of the construct for OX40 was confirmed by comparing uptake to a ^89^Zr-labelled isotype control antibody. To confirm that the tracer uptake corresponded to the location of the inoculated brain tumor, the PET/CT images were compared to MRI imaging. These results also indicated that the extent of uptake might be dependent on tumor volume. The tracer amount of the probe had no inhibitory effect on the treatment and treatment response could be predicted with the tracer, making this a promising radiopharmaceutical for monitoring clinical cancer immunotherapy strategies [118].

#### 3.1.6. NKp30 (NK Cells)

Natural killer cells are innate lymphoid cells that can kill targets without prior sensitization. NK cells kill cells that do not express the MHC class I molecules and can thus destroy tumor cells that evade T cell detection by downregulation of these molecules. NK cell-based therapies consist of adoptive cell transfer (CAR NK cells) or modulating NK cell response in situ. A ^99m^Tc-radiolabelled antibody that images neural cell adhesion molecule (NCAM, CD56), expressed on NK cells, has already been developed. However, NCAM is expressed on many types of cells, such as tumor cells, T cells, dendritic cells and monocytes, which decreases tracer specificity [25].

Therefore, Shaffer et al. developed two immunoPET tracers that bind a natural cytotoxicity receptor expressed on NK cells, NKp30, which induces a strong cytotoxic effect when binding to its tumor ligand B7-H6 [119]. [^64^Cu]Cu-NKp30Ab was synthesized by labelling DOTA-derived NKp30Ab with ^64^Cu, whereas [^89^Zr]Zr-NKp30Ab made use of DFO as a chelator. [^64^Cu]Cu-NKp30Ab and [^89^Zr]Zr-NKp30Ab showed good in vitro immunoreactivity of 72.3% and 63.8%, respectively. [^64^Cu]Cu-NKp30Ab demonstrated specific in vitro binding to NK92MI cell line and primary human NK cells isolated from buffy coats. In vivo PET/CT imaging showed specific uptake of [^64^Cu]Cu-NKp30Ab in NKp30-expressing Hela tumor xenografts (15.2 ± 4.5 %ID/g) in comparison to NKp30-negative wild-type Hela-controls (5.8 ± 1.9 %ID/g). At the terminal time point (48 h post injection) the amount of [^64^Cu]Cu-NKp30Ab in the blood was still high (8.3% ID/g). In contrast, [^89^Zr]Zr-NKp30Ab showed an improved blood to tumor ratio at 120 h post injection, demonstrating that the longer half-life of ^89^Zr (78.4 h) aligns better with the long circulation time of mAbs [25].

### 3.2. Imaging of Metabolic Markers

The activation of T cells is accompanied by metabolic reprogramming, making use of adaptive metabolic pathways [120,121]. [^18^F]FDG has been explored to quantify immune response by measuring the increased glycolysis rate. However, due to non-specific uptake in all GLUT1 or GLUT3 expressing cells and increased uptake in any cell, due to inflammation, [^18^F]FDG is unsuitable for assessing immune cell involvement in immunotherapy [122]. An alternative method to monitor immune response imaging metabolic reprogramming, is to image DNA salvage pathways. By imaging specific enzymes of these metabolic pathways, a distinction can be made between activated and non-activated T cells [93]. Thymidine kinase 1 (TK1), deoxycytidine kinase (dCK) and deoxyguanosine kinase (dGK) have been evaluated as potential imaging targets by different groups due to their role in the DNA salvage pathway [93,123,124]. 

The radiotracers described here aim to target a metabolic process that is specific for T cells. After tracer injection, it is taken up by T cells due to its unique metabolic reactivity in these cells. 

#### 3.2.1. TK1

Thymidine kinase (TK) 1 is a DNA salvage pathway enzyme, which is upregulated during lymphocyte proliferation and is therefore a suitable target for imaging T cell activation [124]. After thymidine enters the cell, TK1 phosphorylates it to its monophosphate, whereafter it is converted to its triphosphate form ready for DNA synthesis [124]. [^18^F]FLT, a thymidine analog targeting DNA synthesis was developed to image cell proliferation [57]. [^18^F]FLT enters the cell through passive diffusion and carrier-mediated transport, and after phosphorylation by TK1, is trapped in the cell [125,126]. This radiotracer can be used to monitor lymphocyte activation in the lymph nodes, in good agreement with T cell proliferation, which was not observed with [^18^F]FDG [57]. However, it has been proven that TK1 levels are also increased in different types of cancers, such as lung, colon, breast and prostate, thus making imaging of immune cell activation at the tumor site unreliable [124]. Therefore, measurements of TK1 as a biomarker for immune activation are more relevant in the lymphatic system.

Aarntzen et al. measured [^18^F]FLT PET signal in patient lymph nodes after vaccination with antigen-loaded dendritic cells [57]. The reported results demonstrate that [^18^F]FLT accumulation correlates well with activated lymphocyte proliferation, and that [^18^F]FLT can be used to determine therapy response [57]. Ribas et al. evaluated [^18^F]FLT uptake in melanoma patients after treatment with tremelimumab, an anti-CTLA-4 antibody [28]. The [^18^F]FLT uptake in the spleen was increased after treatment, whereas the [^18^F]FDG spleen uptake did not change significantly. However, pre- and post-treatment scans at the tumor site did not differ significantly for either [^18^F]FLT or [^18^F]FDG, indicating that treatment response cannot be measured at the tumor site [28]. The PET scans to determine immunoactivation were done between one and two months after treatment, which is earlier than the five months’ median time to determine tumor response by CT scan [28]. Scarpelli et al. analysed [^18^F]FLT uptake after an intradermal injection of pTVG-HP vaccine in patients [94]. The results showed an increased uptake in draining lymph nodes compared to non-draining nodes. Another arm of the study received a combination of pTVG-HP vaccine and pembrolizumab. The splenic [^18^F]FLT uptake was the highest in this arm. However, increased splenic uptake was inversely correlated with progression-free survival time making interpretation of the results challenging. We can thus conclude that [^18^F]FLT will not work in the clinic for this purpose. Interestingly, researchers also noted that baseline FLT uptake in the thyroid was predictive for the occurrence of thyroid-related adverse events [94].

#### 3.2.2. DCK

Deoxycytidine kinase (dCK) is a rate-limiting enzyme in the deoxyribonucleoside salvage pathway that is upregulated during lymphocyte proliferation and is therefore suitable for T cell activation imaging. After a deoxyribonucleoside enters the cell, it is phosphorylated to its monophosphate by dCK, whereafter its triphosphate form is incorporated into DNA during DNA replication [127].

1-(2′-deoxy-2′-[^18^F]fluoroarabinofuranosyl) cytosine ([^18^F]FAC), is a deoxycytidine analog that is predominantly incorporated into the salvage pathway for DNA synthesis in lymphoid organs and rapidly proliferating tissues. This tracer was developed as a high-affinity substrate for dCK to image immune activation in secondary lymphoid organs and is trapped inside the cell after phosphorylation [93]. It has been demonstrated that the uptake of this tracer is increased in activated CD8^+^ T cells compared to naïve T cells [93]. The tracer was produced with high chemical and radiochemical purity with late-stage radio-fluorination of the pyrimidine. However, [^18^F]FAC can be deaminated in vivo by cytidine deaminase [128,129]. The biodistribution data in C57/BL6 mice showed that there is a preferential retention in high dCK-expressing tissue, such as lymphocytes, bone-marrow cells and enterocytes. Using an onco-retrovirus tumor model in mice, increased dCK activity, due to T cell activation, could be measured. The results showed that at day 15 after injection, there was increased activity in the spleen and tumor draining lymph nodes and that effector CD8^+^ T cells had a four times higher retention of [^18^F]FAC than naïve T cells. [^18^F]FAC is more selective for lymphoid organs than [^18^F]FDG and [^18^F]FLT; however, the baseline retention in lymphoid organs is higher and could hinder the detection of weak immune responses [93]. Additionally, Chen et al. demonstrated that activated leukocytes labelled with [^18^F]FAC can cross the blood-brain-barrier and be used to visualize leukocyte accumulation in the brain using an experimental autoimmune encephalomyelitis model [130]. A drawback of this tracer is the long synthesis time and rapid in vivo deamination of [^18^F]FAC, making it less suitable for regular use in the clinic [131]. 

[^18^F]FAC analogs L-[^18^F]FMAC and L-[^18^F]FAC have similar specificity to dCK as [^18^F]FAC but higher metabolic stability than [^18^F]FAC and have reached the clinic [128,132]. Unfortunately, both tracers have a high uptake in the myocardium due to reactivity with TK2, which leads to high background uptake [128].

Another substrate of dCK, [^18^F]Clofarabine ([^18^F]CFA), was developed by Kim and colleagues for imaging lymphocyte activation in secondary lymphoid organs [133]. Unlike [^18^F]FAC, it is not a good substrate of TK2 and has high in vivo stability, making it a more suitable tracer to image dCK selectively. However, deoxycytidine (dC), which has a higher blood concentration in mice than in humans, is a competitive inhibitor of dCK, which makes immediate comparison between mice and humans impossible [133]. A first in human study showed that the tracer is cleared renally and that it distributes to dCK positive tissues (bone marrow, liver, spleen and axillary lymph nodes) of healthy volunteers [133]. 

#### 3.2.3. DGK

Arabinofuranosylguanine (AraG), a modified sugar specifically cytotoxic for T lymphoblasts, is a substrate of mitochondrial deoxyguanosine kinase (dGk) that is upregulated in activated T cells [123]. Namavari et al. have adapted AraG to a PET-tracer, [^18^F]F-AraG, to visualize T lymphoblast and T lymphocyte immune activation and proliferation in cancer [123]. The group was able to develop [^18^F]F-AraG with a radiochemical, decay corrected, yield of 7–10% and a specific activity of 0.8–1.3 Ci/µM by direct fluorination of a trifylate precursor. Uptake in activated T cells was confirmed using in vitro assays in the CCRF-CEM cell line and primary murine T cells [123]. Ronald et al. investigated [^18^F]F-AraG imaging in the context of acute graft-versus-host disease due to allogenic cell transplant and demonstrated the T lymphocyte specific uptake of [^18^F]F-AraG in different immune cell lines and primary T cells isolated from Balb/C mice [134]. Levi et al. were able to predict treatment response to anti-PD-1 therapy in MC38 tumor-bearing mice using [^18^F]F-AraG imaging 48 h after treatment [135]. A first in human [^18^F]F-AraG biodistribution study was performed in six healthy volunteers which showed tracer accumulation in kidneys, liver and heart and proved the safety of [^18^F]F-AraG at tracer concentration [134]. Recently, the ability of [^18^F]F-AraG to visualize T cell infiltration and inflammatory demyelisation and to evaluate lesion heterogeneity in a multiple sclerosis model in mice was proven [136]. The NCT03311672 clinical trial that examined [^18^F]F-AraG accumulation in patients with lung cancer treated with pembrolizumab and either with, or without, radiation therapy was stopped due to low tracer uptake. However, a high number of clinical trials are being performed with this tracer in different stages for different cancers and viral diseases. Of those, the NCT04726215 trial that will enrol about 50 non-small cell lung cancer (NSCLC) patients to image T cell activation before and after anti-PD-1/PD-L1 therapy is the largest. 

### 3.3. Imaging of Exhaustion Markers

Currently, ICI, that target exhaustion markers on immune cells and tumor cells, are the backbone of immunotherapy in oncology. Exhaustion markers, such as PD-1, CTLA-4 and LAG3, are transiently expressed on active immune cells. However, the co-expression of multiple exhaustion markers is characteristic for immune cell dysfunction [137]. To increase the percentage of successful tumor remission, these markers can be targeted using specific radiotracers and imaged, giving clinicians an indication of their expression levels, which can be used to predict the effectiveness of immunotherapy. In the following paragraphs, radiotracers for exhaustion markers on immune cells are discussed. 

#### 3.3.1. PD-1

Programmed cell-death protein 1 (PD-1) is an inhibitory receptor that is expressed mostly on the cell surface of T cells and NK cells [138]. Different ICI have been developed and approved to block PD-1/PD-L1 interaction and multiple clinical trials are still ongoing [139]. All the approved anti-PD-1 ICIs (pembrolizumab, nivolumab, cemiplimab and dostarlimab) are full-length antibodies and the developed radiopharmaceuticals are derived thereof [140,141].

Hettich et al. developed a [^64^Cu]Cu-NOTA-PD-1 mice mAb which showed specific uptake in lymph nodes and spleen and was able to visualize PD-1^+^ T cells tumor infiltration in a murine model treated with radiotherapy and CTLA-4/PD-L1 inhibitors [26]. Natarajan and colleagues developed two PET tracers, [^64^Cu]Cu-DOTA-pembrolizumab and [^89^Zr]Zr-DFO-pembrolizumab, with good immunoreactivity (74% and 72%, respectively) and tested them in a humanized NSG mouse model engrafted with hPBMCs isolated from human blood. [^64^Cu]Cu-DOTA-pembrolizumab showed a higher liver uptake and liver clearance time than [^89^Zr]Zr-DFO-pembrolizumab, however, at 24 hours after injection the tumor-to-blood ratio was three times higher for [^64^Cu]Cu-DOTA-pembrolizumab than for [^89^Zr]Zr-DFO-pembrolizumab, potentially due to higher chelation stability, making the former a more promising tracer [142].

This year, Kok and colleagues administered [^89^Zr]Zr-pembrolizumab to 7 NSCLC and 11 melanoma patients before and after ICI treatment. The tracer was safe and accumulated slowly in the spleen and tumor site with a maximum at day 7. Tumor uptake correlated with survival and progression free survival of the patients. Interestingly, tracer accumulation was also noted at other inflammation sites [143]. A previous clinical study performed by Niemeijer and colleagues administering [^89^Zr]Zr-pembrolizumab in 12 NSCLC patients had one grade 3 adverse event (myalgia). However, the tracer is still considered safe. In this study, no statistically significant correlation could be made between tracer uptake and treatment response, although a similar trend was reported by the researchers [144].

England and colleagues developed [^89^Zr]Zr-DFO-nivolumab that was able to bind PD-1^+^ T cells in vivo in the tumor and salivary glands of a murine model with a maximum uptake at 168 hours post-injection [145]. Cole and colleagues analysed the biodistribution of [^89^Zr]Zr-DFO-nivolumab in three non-human primates and noted tracer accumulation in the spleen, which was tremendously reduced after co-injection with unlabelled nivolumab, indicating specific binding to PD-1^+^ T cells [146]. Unfortunately, bone accumulation was observed, due to the release of ^89^Zr from the unstable complex formed with DFO, which should be solved by using next generation chelators resulting in a complex with increased stability [146].

Recently, [^89^Zr]Zr-nivolumab was administered to 13 patients with advanced NSCLC to evaluate biodistribution and safety. The results showed that (1) the tracer is safe; (2) the radiopharmaceutical is mainly excreted through the GI tract; (3) there is a high uptake in the spleen; and (4) that the tumor-to-background ratio is high enough for imaging. The researchers were able to correlate radiotracer uptake with PD-1^+^ immune cell aggregates in the tumors and with nivolumab therapy response in lesions larger than 20 mm. However, as expected, the tracer distribution from blood to target is slow and the optimal measuring time point is 5–7 days after injection, which complicates clinical application [147]. 

Multiple PD-L1 targeting tracers have been developed and assessed by different research groups to predict and assess treatment response [7]. 

#### 3.3.2. CTLA-4

Cytotoxic T lymphocyte-associated antigen-4 (CTLA-4), also known as CD152, is a membrane protein expressed on CD4^+^ and CD8^+^ T cells that is involved in the downregulation of T cell immune response at the tumor site upon interaction with CD80/B7.1 and CD86/B7.2 on APC [91,148]. Importantly, CTLA-4 is also expressed in tumor cells [148]. CTLA-4 blockade therapy with the mAb ipilimumab and tremelimumab were developed, and ipilimumab has received FDA approval for treating advanced melanoma and other cancers in combination therapies [91]. Therefore, tracers targeting CTLA-4 have been developed to distinguish between treatment responders and non-responders and an ongoing clinical study performed with the full length mAb [^89^Zr]Zr-ipilimumab in patients diagnosed with metastatic melanoma has already shown promising preliminary results (NCT03313323) [149]. 

Higashikawa et al. developed a PET probe targeting murine CTLA-4, [^64^Cu]Cu-DOTA-anti-mouse CTLA-4 mAb, that showed 86.3 (±2.8%) immunoreactivity of the original CTLA-4 mAb. The results of PET and ex vivo biodistribution studies in immune deficient BALB/c mice grafted with syngeneic mouse tumor model (CT26), which expresses CTLA-4, indicated specific uptake of [^64^Cu]Cu-DOTA-anti-mouse CTLA-4 mAb when compared to the IgG control mAb [91]. Ehlerding and colleagues developed three different ipilimumab-based immunoPET tracers to image human CTLA-4. [^64^Cu]Cu-DOTA-ipilimumab, [^64^Cu]Cu-NOTA-ipilimumab and the antibody fragment [^64^Cu]Cu-NOTA-ipilimumab-F(ab’)_2_, all showing preserved immunoreactive fractions, were assessed in vivo in a NSCLC humanized mouse model. The use of NOTA instead of DOTA in the latter radiopharmaceuticals was motivated by an enhanced in vivo kinetic stability of the Cu^II^-NOTA complexes [150]. Increased radiotracer accumulation of [^64^Cu]Cu-NOTA-ipilimumab was observed in human T cells in the murine models when compared to the antibody fragments; however, higher contrast was measured for the fragment, due to faster clearance [151]. At the moment, no clinical trials are being performed with any of these ^64^Cu-based tracers.

In an attempt to improve the clinical benefit of PD-1 blockade, Mazor and colleagues have developed a monovalent bispecific PD-1/CTLA-4 antibody, MEDI5752 [152]. MEDI5752 was radio-labelled with ^89^Zr to study the activity in vivo of this antibody using a transgenic mouse model that expresses human PD-1 and CTLA-4 on immune cells. They showed that the radiotracer accumulation in the tumor was higher when compared with a conventional anti-CTLA-4 mAb, but not with anti-PD-1 mAb [152]. This suggests that the biodistribution of MEDI5752 was dependent on PD-1 expression and binding. Different clinical trials with un-labelled bispecific mAb are already ongoing.

#### 3.3.3. LAG-3

Recently, lymphocyte-activation gene 3 (LAG-3) has been identified as an immune checkpoint expressed on the surface of CD4^+^ and CD8^+^ T cells, NK cells, dendritic cells, regulatory T cells, B cells and macrophages [27]. Relatlimab was the first blocking antibody to be developed and is currently undergoing Phase II/III trials [27]. Recently, the combination of relatlimab with nivolumab was shown to be superior to nivolumab alone in preventing disease progression or death in patients with untreated metastatic or unresectable melanoma, making the development of a diagnostic tool for LAG-3 of value [153,154]. 

Therefore, Lecocq and colleagues developed a ^99m^Tc-labeled single-domain antibody and evaluated it via SPECT/CT in tumor-bearing mice [27]. The tracer accumulated in the kidneys, bladder, and tumors at 80 minutes post-injection. An increased tracer uptake was observed in lymph nodes and tumor when mice were treated with PD-1 inhibitors, due to increased LAG-3 expression on TILs [27]. A PET tracer using ^89^Zr labelled to the human antibody REGN3767 with the bifunctional chelator DFO was developed by Kelly and colleagues and demonstrated high immunoreactivity in cell binding assays [155]. Tracer specificity for LAG-3 was confirmed in tumors expressing human LAG-3 in immune deficient mice by comparing uptake with a ^89^Zr-isotype control antibody. Furthermore, the ability to label intra-tumoral and splenic activated T cells was assessed by implanting Raji lymphoma cells with human peripheral blood mononuclear cells [155]. At the moment, interpretation of the results is difficult and the high accumulation in the liver is a nuisance. Additionally, the optimal timepoint for tracer administration has to be investigated. This tracer is currently under clinical investigation (NCT04566978).

## 4. Conclusions and Future Perspectives

In conclusion, molecular imaging of cytotoxic immune cells is a valuable tool to predict and evaluate immunotherapy treatment response and to aid clinical decision-making for treatment of tumors. However, each of the different cell-imaging strategies has strengths that can be exploited and weaknesses that must be mitigated. The value of direct in vivo imaging of lineage markers such as CD8 for CD8^+^ T cell accumulation at the tumor site has already been demonstrated and clinical studies with [^89^Zr]Zr-DFO-IAB22M2C are being performed [21,77]. However, in our opinion, tracers that target activation or exhaustion markers and give information on the functional state of the cells in addition to the immune cell infiltration in the tumor are the most promising, as these tracers can provide clinicians with more comprehensive information on (potential) treatment response. The high predictive value of [^68^Ga]Ga-NOTA-GZP in CT26 and MC38 colon cancer mice models further substantiates this claim [98]. However, some strategies have demonstrated unclear results, hindering the clinical use of the tracers, as seen for IL-2R targeting probes [110]. As of now, multiple tracers are being developed to target activation/exhaustion markers and several clinical studies are ongoing.

However, there are still opportunities to develop tracers for other activation/exhaustion markers. One of the most promising exhaustion targets, T cell immunoglobulin and mucin domain-3 (TIM3), is upregulated on CD4^+^ and CD8^+^ T cells after cell activation and is expressed on exhausted NK cells as well [156,157]. Immune cell death is induced after binding of TIM3 to its ligand galectin-9, thus inhibiting the immune response [156]. Multiple anti-TIM3 clinical trials are already ongoing, and therefore, it will be of value to be able to discriminate between patients that would benefit from this therapy early on [156]. Another exhaustion marker of interest is the T cell immunoreceptor with immunoglobulin and ITIM domain (TIGIT), that is expressed on both activated T cells and NK cells, and promotes immune cell exhaustion through multiple pathways [158]. Different therapies are being developed targeting TIGIT or its ligands and clinical trials are ongoing [158]. Similar to other exhaustion markers, a diagnostic tool would be of value to improve treatment efficacy once used in the clinic. A lot of research is being done in promoting the effector function of NK cells because of their ability to recognise and lyse MHC class I-deficient cells. Different targets, such as NKG2A and CD96, are being evaluated as therapeutic targets and diagnostic markers will be necessary in the future [157]. Another target of interest on NK cells is the NKp46 receptor, which is only expressed on NK cells, irrelevant of their functional status. The development of a NKp46-specific tracer could further elucidate the role of NK cells in immune response to cancer [159].

Conventionally, radio-labelled mAbs have been used to image targets in vivo due to their high specificity and, in some cases, their corresponding therapeutic usage. However, the large size of full-length antibodies limits their tumor penetration and the accompanying long half-life in the bloodstream decreases contrast due to high background activity [76]. Therefore, a trend seen in the development of new radiopharmaceuticals is to use smaller vectors. The smaller antigen-binding fragments (Fv) suffer lower stability and target affinity than antibodies, which may limit their usage [160]. The heavy chain-only antibodies from camelids, nanobodies, are being used more often because the affinity for antigens is sufficient and their molecular weight is only 12-15 kDa, allowing renal elimination [161]. Examples of this tendency are demonstrated in the development of CD8^+^ T cell-targeting probes (Table 3). Adnectins, even smaller constructs (10 kDa), have recently been explored for PET imaging of PD-L1 (^18^F-BMS-986192, ^68^Ga-BMS-986192) and have shown promising (pre-)clinical results [147,162,163]. All these results point towards the high potential of developing smaller constructs in immunoPET, increasing the imaging contrast and limiting the radiation burden for patients.

To obtain an optimal target/background ratio, the half-life of the radionuclide should approximate the plasma half-life of the tracer [164]. As the plasma half-life of peptides and antibody fragments is short, suitable radionuclides, such as ^18^F (t_1/2_ = 109.8 min) and ^68^Ga (t_1/2_ = 67.8 min), have to be used [164,165]. The many advances (e.g., late-stage radiofluorination) make the development of different ^18^F-and ^68^Ga-probes possible [166]. Another approach being investigated to reduce the dose delivered to patients, while retaining the high affinity of antibodies, is the use of pre-targeting strategies. These strategies consist of radionuclide administration a few days after administration of a modified antibody, making use of specific biorthogonal reactions, such as strain-promoted azide-alkyne cycloaddition, to couple the high antibody affinity with the favorable kinetic properties of small imaging probes [167,168]. 

To conclude, significant advances have been made in the molecular imaging of effector immune cells for evaluation of cancer treatment. Still, many challenges and opportunities exist to routinely use these in a clinical setting.

## Figures and Tables

**Figure 1 biomedicines-10-01074-f001:**
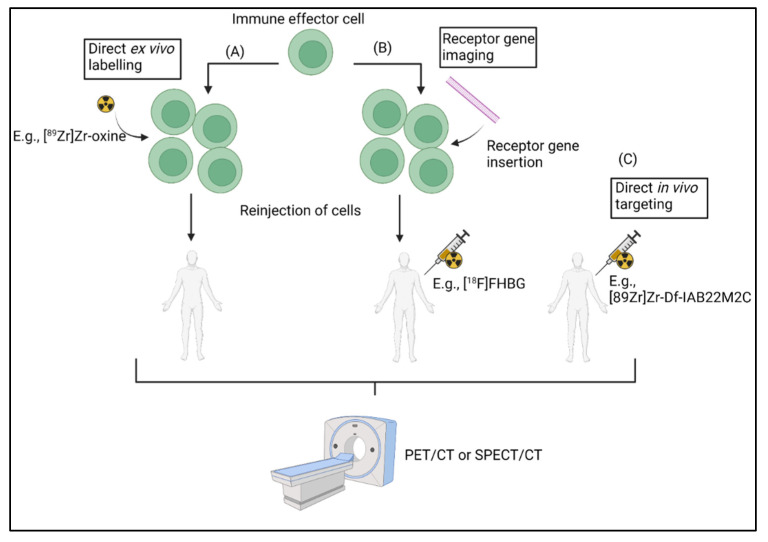
Schematic illustration representing the different methods to image immune cells with PET and SPECT radiopharmaceuticals. (**A**) direct ex vivo labelling, (**B**) receptor gene imaging and (**C**) direct in vivo targeting. Created with Biorender.com.

**Figure 2 biomedicines-10-01074-f002:**
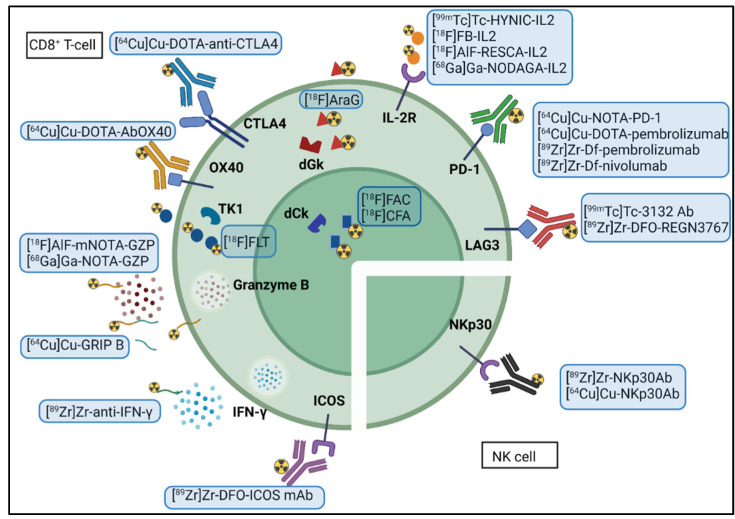
Tracers that image cytotoxic immune cells’ functional state. Created with BioRender.com.

**Table 1 biomedicines-10-01074-t001:** Tracers and targets of the direct ex vivo labelling method used in adoptive cell transfer.

Direct Ex Vivo Labelling			
Target	Tracer	Type	Stage
Metabolic tracer	[^18^F]FDG	Small molecule [39,41]	Clinical
Intracellular trapping	[^89^Zr]Zr-oxine	Small molecule [42,43]	Clinical
Intracellular trapping	[^111^In]In-oxine	Small molecule [39,44]	Clinical
Cell surface proteins	[^89^Zr]Zr-DFO-Bz-NCS	Small molecule [45]	Preclinical
Intracellular trapping	[^64^Cu]Cu-PTSM	Small molecule [46,47]	Preclinical

**Table 2 biomedicines-10-01074-t002:** Tracers and targets of the reporter gene method used in adoptive cell transfer.

Reporter Gene Strategy			
Reporter Gene	Tracer	Type	Stage
*HSV1-TK*	[^18^F]FHBG	Small molecule [50,51]	Clinical (NCT00185848)
	[^18^F]FIAU	Small molecule [52,53]	Preclinical
	[^18^F]FEAU	Small molecule [54]	Preclinical
	[^18^F]FHPG	Small molecule [55,56]	Preclinical
	[^18^F]FLT	Small molecule [57,58]	Clinical (e.g., NCT00585741)
	[^18^F]FMAU	Small molecule [59,60]	Clinical (e.g., NCT04752267)
	[^124^I]FIAU	Small molecule [61,62]	Clinical (NCT01337466)
*hdCKDM*	[^18^F]FEAU	Small molecule [54]	Preclinical
	[^124^I]FIAU	Small molecule [61,62]	Clinical (NCT01337466)
	L-[^18^F]FMAU	Small molecule [63,64]	Preclinical
Sodium iodide symporter (*NIS*)	[^99m^Tc]pertechnate	Small molecule [32,65]	Clinical (e.g., NCT04563780)
	[^18^F]FTB	Small molecule [66]	Preclinical
Somatostatin receptor 2 (*SSTR2*)	[^18^F]AlF-NOTA-octreotide	Small molecule [67]	Clinical (e.g., NCT04552847)
	[^68^Ga]Ga-DOTATOC	Small molecule [68]	Clinical (e.g., NCT02359500)
Norepinephrine transporter (*NET*)	[^18^F]MFBG	Small molecule [69]	Clinical (e.g., NCT04258592)
	[^124^I]MIBG	Small molecule [69]	Clinical (e.g., NCT01583842)
*eDHFR*	[^11^C]TMP	Small molecule [70,71]	Preclinical
	[^18^F]F-TMP	Small molecule [72]	Clinical (NCT04263792)
*PSMA*	[^18^F]DCFPyL	ABP [33]	Clinical (NCT03424525)
	[^18^F]F-DCFBC	Small molecule [73]	Clinical (e.g., NCT01815515)
	[^18^F]PSMA-1007	peptide [74,75]	Clinical (e.g., NCT04487847)

**Table 3 biomedicines-10-01074-t003:** Tracers and targets of the direct in vivo labelling method.

Direct In Vivo Labelling			
Target	Tracer	Type	Stage
CD8	[^89^Zr]Zr-PEG20-x118-VHH	Heavy chain only IgG (VHH) [77]	Preclinical
	[^64^Cu]Cu-NOTA-2.43	Minibody (Mb) [78,79]	Preclinical
	[^64^Cu]Cu-NOTA-YTS169	Mb [78,80]	Preclinical
	[^89^Zr]Zr-malDFO-169	Mb [80,81]	Preclinical
	[^89^Zr]Zr-malDFO-2.43	Bivalent antibody fragment (cDb) [79,82]	Preclinical
	[^89^Zr]Zr-DFO-IAB22M2C	Mb [83]	Clinical (NCT03802123, NCT03107663, NCT04955262)
VLA-4	[^64^Cu]Cu-LLP2A	Peptide [84,85]	Preclinical
TCR	[^89^Zr]Zr-DFO-aTCRmu-F(ab’)2	Antibody (Ab) [34,86]	Preclinical
	[^64^Cu]Cu-DOTA-cOVA-TCR	mAb [87,88]	Preclinical
HLA-DR	^64^Cu-labelled VHH	VHH [89]	Preclinical
NCAM (=CD56) (NK cells)	^99m^Tc-labelled anti-CD56 mAb	mAb [90]	preclinical

## Data Availability

Not applicable.

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
