# Peer review of "Radionuclide Imaging of Cytotoxic Immune Cell Responses to Anti-Cancer Immunotherapy"

_biomedicines, 2022, doi:10.3390/biomedicines10051074_

Round 1
Reviewer 1 Report
1- Important results must be included in both abstract and conclusion.
2- Reference are more, satisfy with the recent ones.
3- Revise the language of all manuscript.
Author Response
Below are point-by-point responses for the reviewers’ comments.
1- Important results must be included in both abstract and conclusion.
Reply: Most important results were added in line 25-32 of the abstract and also incorporated in the conclusion.
2- Reference are more, satisfy with the recent ones.
Reply: More recent references were added to complement older ones.
3- Revise the language of all manuscript.
Reply: Correction of grammatical errors and English improvement were carried out as suggested.
Reviewer 2 Report
The manuscript entitled "Radionuclide imaging of cytotoxic immune cell responses to anti-cancer immunotherapy" submitted by Louis L et al. complies a review of present and past non invasive PET radionuclide imaging of cytotoxic immune responses to immunotherapy. Please see comments below:
1) As a minor comment, author affiliations needs to be bigger in font.
2) Please proof read manuscript thoroughly as there were repetitive words and typos, missing spaces, for example, line 89 has a repetitive word. please correct them and check through entire manuscript.
3) In table 1, the authors have mentioned about cell surface protein tracer as [89Zr]Zr-DFO-Bz-NCS for direct ex vivo labeling, please describe the tracers metabolic stability in vivo and ex vivo?
4) In line 220, the authors have used IETD and IEPD in the manuscript, please discuss the terms and discuss how they are highly specific to human granzyme?
5) Please also discuss or include immunoreactivity of some or all the mentioned PET tracers whether their metabolic stability is suitable for cytotoxic immune cells?
Address as minor revision
Author Response
1) As a minor comment, author affiliations needs to be bigger in font.
Reply: The font size is pre-defined by the journal.
2) Please proof read manuscript thoroughly as there were repetitive words and typos, missing spaces, for example, line 89 has a repetitive word. please correct them and check through entire manuscript.
Reply: The manuscript was scrutinized thoroughly for mistakes and typos.
3) In table 1, the authors have mentioned about cell surface protein tracer as [89Zr]Zr-DFO-Bz-NCS for direct ex vivo labeling, please describe the tracers metabolic stability in vivo and ex vivo?
Reply: The information regarding metabolic stability of this tracer was added to line 137-144: “This efflux is partially mitigated by a more recent tracer, [89Zr]Zr-DFO-Bz-NCS, which forms a biostable covalent thiourea bond between the amine functionalities and the NCS group. However, this radiotracer suffers from the known issue of instability of the 89Zr-DFO metal complex. In order to increase the stability of 89Zr complexes two DFO chelator analogues have been developed and are currently under clinical evaluation [41,42]. Moreover, toxicity to the isolated cells, patient and manipulator must be taken into account when using the direct ex vivo cell labelling method and loss of cell function must be tested extensively [33].”
4) In line 220, the authors have used IETD and IEPD in the manuscript, please discuss the terms and discuss how they are highly specific to human granzyme?
Reply: A mistake was made when writing the sequences. The sequences were changed and more information was added in lines 230-240: “Although the structure and sequence of human granzyme B and mouse granzyme B are similar, their substrate specificity is different with human granzyme B preferring the tetrapeptide sequences IEPD, IETD or IEQD over the IEFD-sequence that is recognized specifically by mouse granzyme B, as determined by Casciola-Rosen and col-leagues using positional scanning synthetic substrate combinatorial libraries [104]. Therefore, the same group developed an analogous biotinylated probe for human granzyme B by changing one amino acid of the tetrapeptide to create hGZP, which showed high specificity to human granzyme B in human melanoma biopsies [25]. Although the IEPD sequence can be also recognized by other proteases such as caspase-8, the fact that granzyme B is released into the extracellular space from activated T cells, makes this protease preferentially accessible to binding by the imaging agent over intracellular counterparts [100,105].”
5) Please also discuss or include immunoreactivity of some or all the mentioned PET tracers whether their metabolic stability is suitable for cytotoxic immune cells?
Reply: This information was added for multiple tracers in sections 3.1 and 3.3
Lines 360-362: “The researchers demonstrated that the immunoreactivity of the DOTA-AbOX was comparable to the immunoreactivity of the unmodified Ab in an in vitro activated murine T cell-binding experiment [122].”
Lines 538-542: “Natarajan and colleagues developed two PET tracers, [64Cu]Cu-DOTA-pembrolizumab and [89Zr]Zr-DFO-pembrolizumab, with good immunoreactivity (74% and 72% of the original mAb, respectively) and tested them in a humanized NSG mouse model en-grafted with hPBMCs isolated from human blood.”
Lines 595-599: “Ehlerding and colleagues developed three different ipilimumab-based immunoPET tracers to image human CTLA-4. [64Cu]Cu-DOTA-ipilimumab, [64Cu]Cu-NOTA-ipilimumab and the antibody fragment [64Cu]Cu-NOTA-ipilimumab-F(ab’)2, all with similar immunoreactivities as the com-mercial antibody, were assessed in vivo in a NSCLC humanized mouse model.”
Lines 627-630: “A PET tracer using [89Zr]Zr labelled to the human antibody REGN3767 with the bi-functional chelator DFO was developed by Kelly and colleagues and demonstrated high immunoreactivity in cell binding assays [162].”